# Delayed Cutaneous Adverse Reaction of the AstraZeneca COVID-19 Vaccine in a Breastfed Female Infant: A Coincidence or a Rare Effect?

**DOI:** 10.3390/vaccines10040602

**Published:** 2022-04-13

**Authors:** Patrícia Diogo, Gil Correia, João B. Martins, Rui Soares, Paulo J. Palma, João Miguel Santos, Teresa Gonçalves

**Affiliations:** 1Institute of Endodontics, Faculty of Medicine, University of Coimbra, 3000-075 Coimbra, Portugal; ppalma@uc.pt (P.J.P.); jsantos@fmed.uc.pt (J.M.S.); 2Medical Microbiology Research Group, CNC—Center for Neurosciences and Cell Biology, 3004-504 Coimbra, Portugal; gil.rcorreia@gmail.com (G.C.); tmfog@ci.uc.pt (T.G.); 3Departments of Endodontology, Academic Centre for Dentistry, 1081 LA Amsterdam, The Netherlands; j.f.brochadomartins@acta.nl; 4Department of Clinical Pathology Service, Instituto Português de Oncologia de Coimbra Francisco Gentil EPE, 3000-075 Coimbra, Portugal; ruisoares@ipocoimbra.min-saude.pt

**Keywords:** AstraZeneca, breastfeeding, cutaneous reaction, COVID-19 vaccines, lactation

## Abstract

The ChAdOx1 nCoV-19 vector vaccine (Vaxzevria, AstraZeneca, Cambridge, UK) was developed at Oxford University and is considered safe for the administration in lactating mothers. Nevertheless, as a novel vaccine, there are gaps in the knowledge regarding possible adverse events in breastfeeding infants of vaccinated mothers. This case report provides first-time data on a possible delayed, cutaneous, adverse reaction in a breastfed, 16-month-old female infant after the first administration of the AstraZeneca vaccine to her 33-year-old mother. Even though, no clinical adverse effects were observed in the mother, her daughter had a 2-day rash in the lower extremities and face. The infant’s cutaneous rashes might be a coincidental event. However, all skin lesions were analogous to previous descriptions and photographs of dermatologic reactions, which resolved spontaneously with no medical intervention, in people who had been vaccinated with other COVID-19 vaccines. Our aim is that this short report contributes to the enhancement of parental awareness about the possibility of similar skin rashes in breastfed children when the mothers receive a vaccination and the importance of reporting those adverse reactions to the competent authorities.

## 1. Introduction

The vaccination of lactating women with inactivated or recombinant vaccines has been advised as safe to breastfeeding mothers and their infants [1,2,3]. However, as a novel vaccine, there are still gaps in the knowledge. The ChAdOx1 nCoV-19 vaccine (Vaxzevria, AstraZeneca, Cambridge, UK) was developed at Oxford University and consists of a replication-deficient chimpanzee adenoviral vector ChAdOx1, which carries the SARS-CoV-2 structural surface glycoprotein antigen (spike protein; nCoV-19) gene that can enter the cells yet is unable to replicate [4]. While the Pfizer-BioNTech (Comirnaty, BioNTech/Pfizer, BNT162b2, Mainz, Germany) and Moderna vaccines (Moderna, mRNA-1273, Cambridge, MA, USA) are messenger ribonucleic acid (mRNA) vaccines, the AstraZeneca/Oxford vaccine is an adenovirus vector vaccine [1,4]; it contains less ingredients when compared with mRNA vaccines, as shown in Table 1, and requires two doses given four weeks apart [3].

The COVID-19 vaccines were introduced in the market after an emergency use authorization for vaccines, and at that time, no completed clinical trials were available [2]. All vaccines have been authorized under a so-called conditional approval scheme by the European Medicine Agency (EMA) while awaiting further evidence. The World Health Organization (WHO) advised that the Pfizer-BioNTech and AstraZeneca COVID-19 vaccines be offered to breastfeeding women, and the discontinuation of breastfeeding was not recommended [2]. As the ChAdOx1 nCoV-19 recombinant vaccine is not a live-virus vaccine, it is biologically and clinically unlikely to promote a risk to the breastfeeding child. Despite the above collected data, there is still a clear gap in knowledge regarding the possible adverse events on breastfed infants whose mothers have been vaccinated with the COVID-19 vaccines. The current work presents a short report of a 33-year-old breastfeeding mother, vaccinated with the first dose of the AstraZeneca vaccine, who did not reveal clinical, adverse signs or symptoms while her breastfed, 16-month-old female infant had cutaneous adverse reactions (lower extremities and face) two days after the mother’s vaccine administration.

## 2. Case Report

A 33-year-old female was vaccinated with the first dose of the AstraZeneca vaccine; a brief questionnaire was developed, and the mother confirmed that she was breastfeeding a female infant. The only medication regularly taken by the patient was a contraceptive pill, 4 milligrams (mg) of progestin (Slinda^®^, Exeltis, Madrid, Spain). One hour after the vaccine, she took 1000 mg of an analgesic paracetamol pill (Ben-u-ron^®^, Bene Farmacêutica, Lisboa, Portugal). No clinical signs or symptoms were registered on the vaccine day or the following days. Two days after the vaccine administration, the 16-month-old infant developed a fever (38.9 °C) at 11:00 p.m., and a 125 mg paracetamol suppository (Ben-u-ron^®^, Bene Farmacêutica, Lisboa, Portugal) was administered. At that precise moment, an annular plaque, measuring three centimeters in diameter, was observed below the knee on the infant’s right leg, as shown in Figure 1(A.1). This episode was reported immediately to the pediatrician, who advised the parents to stay alert and vigilant due to the mother’s administration of the first dose of the AstraZeneca vaccine. After 12 h, the leg lesion fully recovered, and the infant was apyretic. On the third day, two new leg urticarial lesions were observed, one on the back of the right leg Figure 1(A.2) and the other on the anterior region of the left leg Figure 1(A.3). After 10 h, both spontaneously disappeared. On the fourth day after the vaccine, a cutaneous rash was observed on the infant’s face, with marked uniformly erythematous tender plaques on both the right cheek Figure 1(B.1) and the left cheek Figure 1(B.2). After 11 h, both extemporaneously vanished. On the fifth day after the mother’s first vaccine administration, red-marked targetoid cutaneous plaques were seen on the left buttock and the left leg Figure 1(C.1,C.2). These disappeared after 8 h. On the sixth day after the vaccine, the right buttock and right leg showed punctual red lesions that vanished after 12 h Figure 1(D.1,D.2). Due to the constant spontaneous lesions’ improvement and after a telephone discussion with the pediatrician, skin cultures were not advised. The infant only had a fever once. Diarrhea, snot, cough, and oral lesions were not observed. For caution, the infant did not go to kindergarten until full recovery. Apparently, the cutaneous lesions did not cause irritation or itching, and the infant always had an appetite and ate well. The infant was not given any new types of food, and there were no changes to the laundry routines, with the same detergent brands used. Moreover, there were no signs or symptoms of recent or chronic infections, no associations with physical stimuli, and no contacts with inhaled or touched substances. Seven days after the vaccine administration, no further cutaneous lesions were observed, and the pediatrician reported the situation to the competent authorities. During this period, breastfeeding was not interrupted.

One week before the second vaccine dose, the mother was contacted to decide if she wanted the same brand of the vaccine or another brand. Given that neither local injection-site reactions nor delayed-type hypersensitivity reactions were contraindications to the subsequent vaccination, the mother was encouraged by the committee and the pediatrician to receive the second dose and completed her vaccination course with the AstraZeneca vaccine. The infant continued to be breastfed. The second dose of the AstraZeneca vaccine was administrated (lot number ABW4810). The mother’s medication remained the same, and she took 1000 mg of an analgesic paracetamol pill (Ben-u-ron^®^) one hour after the vaccine. Two days after the second vaccine dose, the 19-month-old infant had a fever (39 °C) at 11:15 p.m., and a 125 mg paracetamol suppository was administrated. No cutaneous lesion or rash recurrence was observed. On the morning of the third day, the infant was apyretic, and no clinical signs or symptoms were observed on that day or on the following days. Again, the mother had no clinical adverse reactions. Breast milk analysis and breast swabbing were not performed. One month after the full vaccination program, a laboratory serological analysis was performed both on the mother and on the 20-month-old infant. The results showed that the mother had 318.60 of binding antibody units per milliliter (BAU/mL) of Abs anti-SARS-CoV2 IgG (RBD/S1) (ELISA). Before the first vaccine, the mother had 5.8 UA/mL of Abs anti-SARS-CoV-2 IgG (anti-SP) (CMIA). A serum tryptase (a marker of mast cell degranulation) was quantified. The mother had 5.45 micrograms per liter (µg/L), and the infant had <3.00 of Abs anti-SARS-CoV2 IgG (RBD/S1) (ELISA) and 3.62 µg/L of serum tryptase. While the mother displayed serum immunity, the infant did not. The tryptase detected in peripheral blood is a proenzyme. When released after mast cell degranulation, it appears in its activated form. As the half-life of circulating tryptase is approximately 120 min, the serum levels obtained correlate with the histamine content and with the severity of the anaphylactic condition. The authors conclude that the collection content in the infant may not have been carried out in a detectable phase.

## 3. Discussion

Until the present, none of the COVID-19 vaccines currently used around the world have been trialed in breastfeeding women. This case report provides first-time data on a possible delayed, cutaneous, adverse reaction in a breastfed female infant after the first administration of the AstraZeneca vaccine to her mother. While human milk is not a vector for severe acute respiratory syndrome coronavirus, the milk contains antibodies that can potentially protect breastfed infants from COVID-19 [5,6]. A prospective cohort study in pregnant and lactating women from the US found that vaccines were well tolerated, and vaccine reactions in these groups were like those in women who were not pregnant or breastfeeding [7]. Antibodies generated in response to the vaccine should protect the breastfeeding women and the breastfed infants [7].

Vaxzevria is a vector-based vaccine that uses a modified chimpanzee DNA adenovirus, which does not generate an immune response to itself but rather to the SARS-CoV-2 spike glycoprotein encoded in its DNA [1]. Only two case reports [8,9] of delayed cutaneous reactions with the ChAdOx1 nCoV-19 vaccine were found, analogous to the ones manifested by the present infant lesions. All lesions observed in the breastfeeding infant were similar to previous descriptions of cutaneous reactions around the injection site in people who had been vaccinated with other COVID-19 vaccines [10,11,12,13,14,15] and with a healthy neonate born from COVID-19-positive mother [16]. The authors state that this is a remarkable fact that deserves attention and discussion among healthcare professionals. Furthermore, those several descriptions and photographs are the main reason as well as triggering point for the presentation of this case report, but it is important to mention that until the present moment, this is the only clinical case report. Within this report, the authors propose an undeniable visual relationship as a significant point between the vaccine administration and the infant’s delayed, cutaneous, adverse reactions as there is no other apparent cause for the rash. Due to the cutaneous manifestation’s clinical history, it is hypothesized that all may correspond to the following: (i) a delayed, cutaneous, adverse reaction to mother’s first-dose vaccine (mainly to the inactive components); (ii) a coincidental event (caused by viral infections, e.g., parvovirus, or other unidentified allergens); (iii) an infant response to the mother’s inflammatory reaction to the vaccine; or (iv) a common immune response directed against the spike RNA or proteins inducing virus-associated skin lesions. Since the infant did not develop humoral immunity, it decreases the likelihood that the cutaneous reactions were mediated by the active vaccine contents. Moreover, since the infant did not present a cutaneous reaction with previous vaccines containing polysorbate (e.g., Prevenar 13^®^) or other prior medications, authors excluded the polysorbate allergy, considering that polysorbate 80 (PS80), the AstraZeneca vaccine’s main component, was studied as potentially allergenic or immunogenic [17,18,19]. 

The infant developed a fever twice, both occurring in the second day after the vaccines. However, after the second maternal vaccine dose, the fever occurred with no associated rash. The importance of this report hinges upon its novelty since no similar cases have been reported, and the authors affirm that its novelty is also its major limitation (*n* = 1). Authors invite researchers to observe this patient group with a great deal of attention. The reaction described in this report was mild and benign and should be documented as a possible adverse effect or as a coincidental event. Parents, scientists, clinical doctors, and healthcare professionals should be aware of the possibility of benign skin rashes in infants while breastfeeding. Moreover, breastfeeding should not be discontinued after vaccination, and all adverse reactions should be reported to the competent authorities. Further investigation shall be performed in breastfeeding infants to prove the possible correlation between the COVID-19 vaccination in mothers and the occurrence of an immune response in infants.

## Figures and Tables

**Figure 1 vaccines-10-00602-f001:**
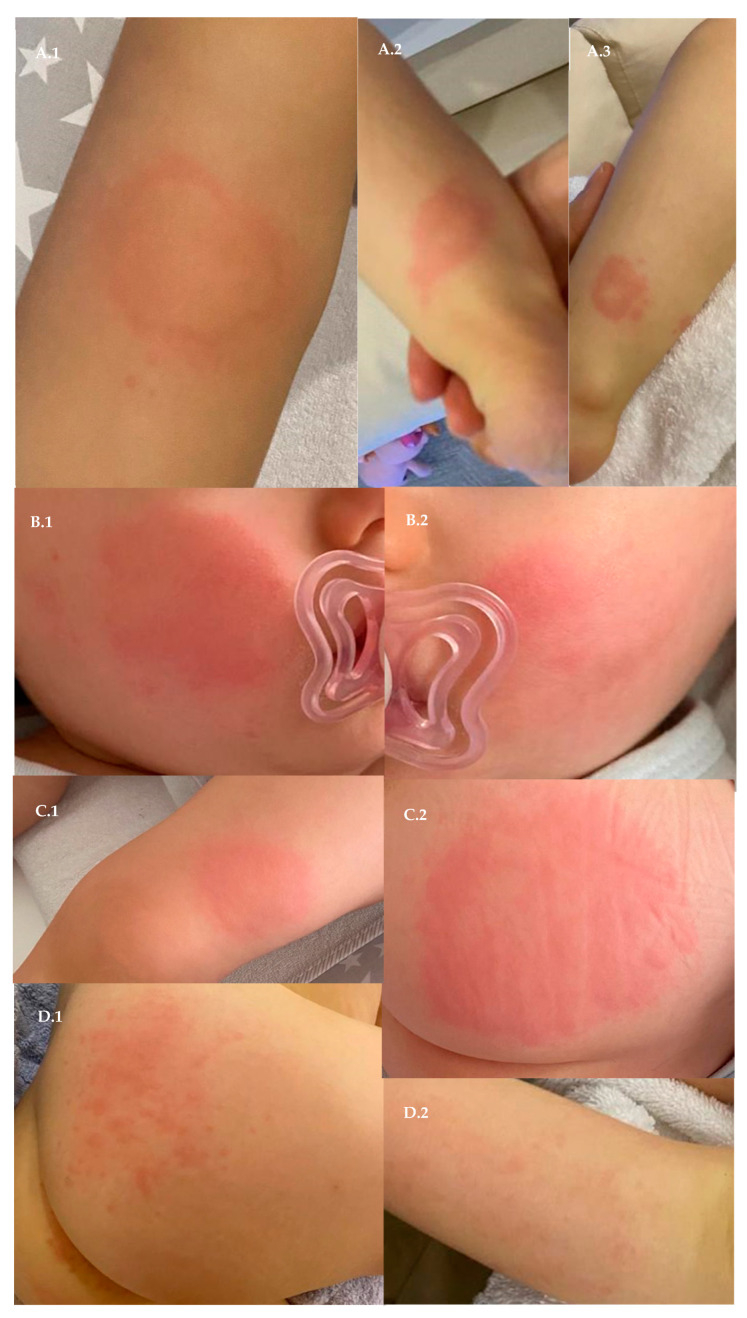
**Cutaneous reaction images in a breastfed, 16-month-old female infant.** Photographs of all lesions are presented from day two until day six after the first dose of the AstraZeneca vaccine. Some photographs were taken using a mirror, so left and right images might be rearranged.

**Table 1 vaccines-10-00602-t001:** Excipients list comparison for AstraZeneca/Oxford^®^, Janssen/Johnson & Johnson^®^, Moderna^®^, and BioNTech-Pfizer^®^ vaccines. Detailed information retrieved from the product characteristic summary available on the website of the European Medicine Agency (EMA) http://www.ema.europa.eu (accessed on 13 January 2022).

AstraZeneca/Oxford^®^ AZD 1222	Janssen/Johnson & Johnson^®^ Ad26.COV2.S	Moderna^®^ mRNA-1273	BioNTech-Pfizer^®^ BNT162b2
Adenovirus vector vaccine (nonreplicating-containing gene, expressing the spike protein)		RNA-based vaccine (mRNA, encoding the spike protein)	
L-histine, histine hydrochloride monohydrate, magnesium chloride hexahydrate, polysorbate 80 (E 433), ethanol, sucrose, sodium chloride, disodium edetate (dihydrate), and water for injections.	2-hydroxypropyl-b-cyclodextrin, citric acid monohydrate, ethanol, hydrochloric acid, polysorbate 80, sodium chloride, sodium hydroxide, trisodium citrate dihydrate, and water for injections.	SM-102 lipid, cholesterol, 1,2 distearoyl-sn-glycero-3-phosphocholine (DSPC), 1,2-Dimyristoyl-rac-glycero-3-methoxypolyethylene glycol-2000 (PEG2000 DMG), tromethamine, tromethamine hydrochloride, acetic acid, sodium acetate trihydrate, and sucrose water for injections.	((4-hydroxybutyl)azanediyl) bis(hexane-6,1-diyl)bis(2-hexyldecanoate) (ALC-0315), 2-((polyethylene glycol)-2000)-N,N-ditetradecylacetamide (ALC-0159), 1,2-distearoyl-sn-glycero-3-phosphocholine (DSPC), cholesterol, potassium chloride, monopotassium phosphate sodium chloride, disodium phosphate dihydrate sucrose, sodium hydroxide, hydrochloric acid, and water for injections.

## Data Availability

Not applicable.

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
