# Peer review of "Delayed Cutaneous Adverse Reaction of the AstraZeneca COVID-19 Vaccine in a Breastfed Female Infant: A Coincidence or a Rare Effect?"

_vaccines, 2022, doi:10.3390/vaccines10040602_

Round 1
Reviewer 1 Report
In this brief case report, Diogo and colleagues present a case of skin rashes and fever in a breastfed infant after the mother received the adenovirus-vectored COVID19 vaccine Vaxzevria.
With this, they provide first-time data on possible adverse effects of vaccination of breastfeeding women on their children. As this is only one case and the conclusions they present is based rather thin evidence, I cannot endorse the report for publication in Vaccines in its current state.
First and foremost, this could be just a coincidence, as the authors also suggest in their title. This should be stressed in the revised manuscript.
Second, no data on blood parameters that point towards an allergic reaction of the child is provided. Neither did the authors take any swabs from mother and child (--> bacteria?) and/or performed breast milk analysis to look for possible sources of the reported rashes. Maybe checking for a polysorbate allergy could be elucidating data?
Next, they compare the rashes in the child to rashes in patients that appeared around the injection site (see references 8 and 9 of the manuscript draft). This is a remarkable fact and discussion on that should be included in the revised manuscript. Skin rashes in infants can have numerous reasons that have to be taken into account. So this really could be a coincidence, especially as n=1.
Additional comments:
- What do the authors refer to with “store the instructions” in lines 39f? Which instructions?
- Reference 1 does not recommend Vaxzevria for lactating women, see line 54f.
- The figure should be “professionalized”, e.g. with letters (a – i) for the different panels and equal image sizes in the rows.
- The authors write that Vaxzevria “uses a technology based on DNA, with inactivated adenovirus genes” – they need to be more specific (“technology based on DNA”, “inactivated adenovirus genes”.
- Line 138f: Reference 16 does not report lesions observed in the breastfeeding, but in a newborn infant. Please revise.
- Line 144: the authors should list all inapparent causes for the rash as differential diagnoses.
- Regarding conclusion i): are polyethylene glycol (PEG) 2000, polysorbate 80 (PS80) and aluminium hydroxide really all ingredients in Vaxzevria? (See line 145f)
- Why do the authors cite ref 19 in line 153?
Author Response
Response to Reviewer 1 Comments
In this brief case report, Diogo and colleagues present a case of skin rashes and fever in a breastfed infant after the mother received the adenovirus-vectored COVID19 vaccine Vaxzevria. With this, they provide first-time data on possible adverse effects of vaccination of breastfeeding women on their children. As this is only one case and the conclusions they present is based rather thin evidence, I cannot endorse the report for publication in Vaccines in its current state. First and foremost, this could be just a coincidence, as the authors also suggest in their title. This should be stressed in the revised manuscript.
Answer: We appreciate the review comments that definitively improve the final quality of the present case report description. This is an important and valid point, that is underlined in the title and the discussion section. Within this, the authors correct the final version (please see highlighted text) in the abstract section and article last paragraph.
Abstract: (…) The infant cutaneous rash might be a coincidental event. However, all skin lesions are analogous to previous descriptions and photographs of dermatologic reactions in people who had been vaccinated with other Covid-19 vaccines resolving spontaneously with no medical intervention.
Line 154: ii) (…) “a coincidental event” (…)
Lines 160-169: “The importance of this report lays on its novelty, since no similar cases have been reported, and the authors affirm that its novelty is also its major limitation (n=1).Within this, authors invite researchers to observe with high attention these patients’ group. The reaction described in this report was mild and benign and should be documented as a possible adverse effect or as a coincidence event. Parents, scientists, clinical doctors, and healthcare professionals should be aware of the possibility of benign skin rashes in infants while breastfeeding. Moreover, breastfeeding should not be discontinued after vaccination and all adverse reactions should be reported to the competent authorities.
Second, no data on blood parameters that point towards an allergic reaction of the child is provided. Neither did the authors take any swabs from mother and child (--> bacteria?) and/or performed breast milk analysis to look for possible sources of the reported rashes. Maybe checking for a polysorbate allergy could be elucidating data?
Answer: We appreciate the observation. However, our understanding has been:
Lines 85-86: “Due to the constant spontaneous lesions improvement, and after telephone discussion with the pediatrician, skin cultures were not advised.”
Concerning the polysorbate allergy, the infant did not present cutaneous reaction with other vaccines containing polysorbate (e.g. Prevenar), neither other previous medication.
Lines 109-111. Mother milk analysis or breast swabs were not performed. One month after the full vaccination program, laboratory serological analysis was performed both in the mother and in the 20-months-old infant.
Next, they compare the rashes in the child to rashes in patients that appeared around the injection site (see references 8 and 9 of the manuscript draft). This is a remarkable fact and discussion on that should be included in the revised manuscript. Skin rashes in infants can have numerous reasons that have to be taken into account. So this really could be a coincidence, especially as n=1.
Answer: This is an important and valid point. Within this, the authors correct the final version (please check the highlighted text).
Abstract: (…) The infant cutaneous rash might be a coincidental event. However, all skin lesions are analogous to previous descriptions and photographs of dermatologic reactions in people who had been vaccinated with other Covid-19 vaccines resolving spontaneously with no medical intervention.
Lines 129-131: “This case report provides first-time data on a possible delayed cutaneous adverse reaction in a breastfed female infant after the administration of the AstraZeneca first vaccine to her mother.”
Lines 141-144: “All lesions observed in the breastfeeding infant were similar to previous descriptions of cutaneous reactions in people who had been vaccinated with other Covid-19 vaccines [10-15] and with a healthy neonate born from Covid-19 positive mother [16].
Lines 161-171: “The importance of this report lays on its novelty, since no similar cases have been reported, and the authors affirm that its novelty is also its major limitation (n=1). Within this, authors invite researchers to observe with high attention these patients’ group. The reaction described in this report was mild and benign and should be documented as a possible adverse effect or as a coincidence event. Parents, scientists, clinical doctors, and healthcare professionals should be aware of the possibility of benign skin rashes in infants while breastfeeding. Moreover, breastfeeding should not be discontinued after vaccination and all adverse reactions should be reported to the competent authorities. Further investigation shall be performed in breastfeeding infants to prove the possible correlation between the Covid-19 vaccination mothers and the occurrence of an immune response in infants.
Additional comments:
- What do the authors refer to with “store the instructions” in lines 39f? Which instructions?
Answer: We appreciate the comment, we believe it is an error. The right expression is “genetic information”. Please see the correct form in the draft (highlighted text).
- Reference 1 does not recommend Vaxzevria for lactating women, see line 54f.
Answer: The reviewer is right. The reference is wrong, please see the correct reference [2] in the draft (highlighted text).
Lines 51-54: “The World Health Organization (WHO), advised that the Pfizer-BioNTech and AstraZeneca Covid-19 vaccines should be offered to breastfeeding women and discontinuing breastfeeding was not recommended [2].”
- The figure should be “professionalized”, e.g. with letters (a – i) for the different panels and equal image sizes in the rows.
Answer: We agree with the reviewer but in the vaccines submission we only could submit the individual photographs and within this we could not introduce letters or number to the individual pictures to have a scientific final image. The images size variation is according to the lesion area to visualize it better.
- The authors write that Vaxzevria “uses a technology based on DNA, with inactivated adenovirus genes” – they need to be more specific (“technology based on DNA”, “inactivated adenovirus genes”.
Answer: The author is totally correct.
Lines 138-140. “Vaxzevria is a vector-based vaccine that uses a modified Chimpanzee DNA adenovirus which does not generate an immune response to itself but rather to the SARS-CoV-2 Spike glycoprotein encoded in its DNA [1].”
- Line 138f: Reference 16 does not report lesions observed in the breastfeeding, but in a newborn infant. Please revise.
Answer: We agree with the reviewer and the correction was made.
Lines 142-145 “All lesions observed in the breastfeeding infant were similar to previous cutaneous descriptions of cutaneous reactions in people who had been vaccinated with other Covid-19 vaccines [10-15] and with a healthy neonate born from Covid-19 positive mother [16].”
- Line 144: the authors should list all inapparent causes for the rash as differential diagnoses.
Answer: We agree with the reviewer and the correction was made.
Lines 150-157:“Due to the cutaneous manifestation’s clinical history, it was hypothesized that all may correspond to: i) delayed cutaneous adverse reaction to mother’s first-dose vaccine (mainly to the inactive components), taking into account that the AstraZeneca’s vaccine main component studied as potentially allergenic or immunogenic is polysorbate 80 (PS80) [17-19]; ii) a coincidental event (caused by viral infections, e.g. Parvovirus; or other unidentified allergens); iii) an infant response to the mother’s inflammatory reaction to the vaccine or iv) a common immune response directed against the spike RNA or proteins inducing virus-associated skin lesions.”
- Regarding conclusion i): are polyethylene glycol (PEG) 2000, polysorbate 80 (PS80) and aluminium hydroxide really all ingredients in Vaxzevria? (See line 145f)
Answer: We agree with the reviewer and the correction was made.
Lines 150-157:“Due to the cutaneous manifestation’s clinical history, it was hypothesized that all may correspond to: i) delayed cutaneous adverse reaction to mother’s first-dose vaccine (mainly to the inactive components), taking into account that the AstraZeneca’s vaccine main component studied as potentially allergenic or immunogenic is polysorbate 80 (PS80) [17-19]; ii) a coincidental event (caused by viral infections, e.g. Parvovirus; or other unidentified allergens); iii) an infant response to the mother’s inflammatory reaction to the vaccine or iv) a common immune response directed against the spike RNA or proteins inducing virus-associated skin lesions.”
- Why do the authors cite ref 19 in line 153?
Answer: We agree with the reviewer and the correction was made.
The reference 19 belongs to references groups in the line 154 [17-19].

Reviewer 2 Report
The authors presented a delayed cutaneous adverse reaction in an infant who was breastfed by a woman 16 months after receiving the first AstraZeneca vaccine. In addition, the woman took the progestin of 4 milligrams (mg) pill. One hour after the vaccine, she took an analgesic paracetamol pill of 1000mg. It is very important that the woman took other pharmaceuticals in addition to the vaccine. Authors should explain it and indicate it in the topic. So, I propose to change the title, e.g. „Delayed cutaneous side effects in a breastfed female infant: pharmacological mix - Covid-19 vaccine, progestin and paracetamol” or something like that
Most COVID-19 candidate vaccines have been developed to induce antispike protein immune responses.
In the other hand AstraZeneca coronavirus disease 2019 (COVID-19) vaccinations have recently been implicated in thromboembolism formations. The most common presenting adverse events were headache, intracerebral hemorrhage, and hemiparesis. The most common thromboembolic adverse events were cerebral venous sinus thrombosis and deep vein thrombosis/pulmonary embolism.
The authors provided a list of excipients for AstraZeneca / Oxford, Janssen / Johnson & Johnson, Moderna and BioNTech-Pfizer vaccines. However, it is advisable that the authors also present the complications of administering Janssen / Johnson & Johnson, Moderna and BioNTech-Pfizer vaccines.
Moreover, the authors did not provide the limitations of the previously presented clinical case.
You should write about limitations and give recommendations
Author Response
Response to Reviewer 2 Comments
The authors presented a delayed cutaneous adverse reaction in an infant who was breastfed by a woman 16 months after receiving the first AstraZeneca vaccine. In addition, the woman took the progestin of 4 milligrams (mg) pill. One hour after the vaccine, she took an analgesic paracetamol pill of 1000mg. It is very important that the woman took other pharmaceuticals in addition to the vaccine. Authors should explain it and indicate it in the topic. So, I propose to change the title, e.g. „Delayed cutaneous side effects in a breastfed female infant: pharmacological mix - Covid-19 vaccine, progestin and paracetamol” or something like that.
Answer: We appreciate the review comments that definitively improve the final quality of the present case report description. In this particular aspect, the progestin pill is regularly taken by the mother and no previous cutaneous rash was observed. In previous occasions, the analgesic paracetamol pill has also been taken with no rash observed. By consulting e-lactancia.org (https://www.e-lactancia.org/breastfeeding/paracetamol/product/) the risk to the baby is very low and no alternatives to this drug are available. Besides, the information that relies on the article is well-founded with the present bibliography with no need to insert more information in the authors opinion.
Lines 142-147: “All lesions observed in the breastfeeding infant were similar to previous descriptions of cutaneous reactions in people who had been vaccinated with other Covid-19 vaccines [10-15] and with a healthy neonate born from Covid-19 positive mother [16]. Those several descriptions and photographs were the main reason as well as the trigger point for the representation of this case report but is important to mention that until the present moment this is the only clinical case report.”
Most COVID-19 candidate vaccines have been developed to induce antispike protein immune responses. In the other hand AstraZeneca coronavirus disease 2019 (COVID-19) vaccinations have recently been implicated in thromboembolism formations. The most common presenting adverse events were headache, intracerebral hemorrhage, and hemiparesis. The most common thromboembolic adverse events were cerebral venous sinus thrombosis and deep vein thrombosis/pulmonary embolism. The authors provided a list of excipients for AstraZeneca / Oxford, Janssen / Johnson & Johnson, Moderna and BioNTech-Pfizer vaccines. However, it is advisable that the authors also present the complications of administering Janssen / Johnson & Johnson, Moderna and BioNTech-Pfizer vaccines.
Answer: This is an advisable comment, but we prefer to answer it, with the politely permission in agreeing to disagree. As authors, we were focused only in two aspects (Covid-19 vaccines cutaneous reactions and excipient lists comparison from Covid-19 vaccines with data from www.ema.europa.eu). Why? Because we waited months with daily inspection in PubMed to find in literature if this rare clinical case (n=1) was similar and/or analogous to others. As we found visual concordance between Astrazeneca vaccine individuals’ cutaneous reactions characteristics with other Covid-19 vaccines, the article was written with bibliography to sustain it. And in the references list, from a total of 19 references, 10 references are articles with cutaneous reactions descriptions. In conclusion, since the vaccine was administered to mother and the reactions are observed in a female infant breastfeeding, the vaccines complications were not considered since the guidelines do not consider the breastfeeding interruption and/or suspension.
Moreover, the authors did not provide the limitations of the previously presented clinical case. You should write about limitations and give recommendations.
Answer: The reviewer is absolutely right, and the modifications were made and described below.
Lines 147-171: “Within this, the authors propose an undeniable visual relationship, as a remarkable fact, between the vaccine administration and the infant delayed cutaneous adverse reactions as there is no other apparent cause for the rash. Due to the cutaneous manifestation’s clinical history, it was hypothesized that all may correspond to: i) delayed cutaneous adverse reaction to mother’s first-dose vaccine (mainly to the inactive components), taking into account that the AstraZeneca’s vaccine main component studied as potentially allergenic or immunogenic is polysorbate 80 (PS80) [17-19]; ii) a coincidental event (caused by viral infections, e.g. Parvovirus; or other unidentified allergens); iii) an infant response to the mother’s inflammatory reaction to the vaccine or iv) a common immune response directed against the spike RNA or proteins inducing virus-associated skin lesions. Since the infant did not develop humoral immunity, it decreases the likelihood that the cutaneous reactions were mediated by the active vaccine contents. The infant had fever twice, both occurring in the second day after the vaccines. However, after the second maternal vaccine dose, the fever occurred with no associated rash. The importance of this report lays on its novelty, since no similar cases have been reported, and the authors affirm that its novelty is also its major limitation (n=1). Within this, authors invite researchers to observe with high attention these patients’ group. The reaction described in this report was mild and benign and should be documented as a possible adverse effect or as a coincidence event. Parents, scientists, clinical doctors, and healthcare professionals should be aware of the possibility of benign skin rashes in infants while breastfeeding. Moreover, breastfeeding should not be discontinued after vaccination and all adverse reactions should be reported to the competent authorities. Further investigation shall be performed in breastfeeding infants to prove the possible correlation between the Covid-19 vaccination mothers and the occurrence of an immune response in infants.”

Round 2
Reviewer 1 Report
The authors have sufficiently addressed all raised issues and improved the manuscript a lot.
However, a few important points should be included in the final version of this report:
- The authors have to mention that they excluded a polysorbate allergy of the infant.
- The authors should really discuss the fact that they compare the rashes in the child to rashes in patients that appeared around the injection site. As stated before, this is a remarkable fact and discussion on that should be included in the revised manuscript.
- The authors write “Unlike the Pfizer-BioNTech (Comirnaty, BioN-38 tech/Pfizer, BNT162b2 Mainz, Germany) and Moderna vaccines (Moderna, mRNA-1273, 39 Massachusetts, EUA) which store the genetic information in single-stranded ribonucleic 40 acid (RNA), the Oxford vaccine uses double-stranded deoxyribonucleic acid (DNA).” -> I suggest they revise these sentences and stress that, while Comirnaty and Moderna are mRNA vaccines, Vaxzevria is an adenovirus vector vaccine. As their sentence suggest in its present form, Vaxzevria could be understood as a DNA vaccine – which is not true.
- Line 64 should state exact information on the vaccine!
- Again, the figure 1 has to be “professionalized”, e.g. with letters (a – i) for the different panels and equal image sizes in the rows. I suggest the authors assemble all individual pictures into an appropriate figure in Adobe AI, PowerPoint, or a similar program. See PMCID: PMC8447195 (Fig. 1) for example…
Author Response
Response to Reviewer 1 Comments
II Version
The authors have sufficiently addressed all raised issues and improved the manuscript a lot.
We appreciate the review comments.
However, a few important points should be included in the final version of this report:
- The authors have to mention that they excluded a polysorbate allergy of the infant.
The reviewer is absolutely right. Changes in the main word were made and are described below.
Lines 156-161: “Since the infant did not develop humoral immunity, it decreases the likelihood that the cutaneous reactions were mediated by the active vaccine contents. Moreover, since the infant did not present cutaneous reaction with previous vaccines containing polysorbate (e.g. Prevenar 13Ò), neither other prior medication, authors excluded the polysorbate allergy (taking into account that the AstraZeneca’s vaccine main component studied as potentially allergenic or immunogenic is polysorbate 80 (PS80) [17-19].”
- The authors should really discuss the fact that they compare the rashes in the child to rashes in patients that appeared around the injection site. As stated before, this is a remarkable fact and discussion on that should be included in the revised manuscript.
Again, the reviewer is correct. Changes in the main word were made and are described below.
Lines 142-149: “All lesions observed in the breastfeeding infant were similar to previous descriptions of cutaneous reactions in people who had been vaccinated with other Covid-19 vaccines around the injection site [10-15] and with a healthy neonate born from Covid-19 positive mother [16]. The authors state that this is a remarkable fact that deserves attention and discussion among healthcare professionals. Furthermore, those several descriptions and photographs were the main reason as well as the trigger point for the representation of this case report but is important to mention that until the present moment this is the only clinical case report.”
- The authors write “Unlike the Pfizer-BioNTech (Comirnaty, BioN-38 tech/Pfizer, BNT162b2 Mainz, Germany) and Moderna vaccines (Moderna, mRNA-1273, 39 Massachusetts, EUA) which store the genetic information in single-stranded ribonucleic 40 acid (RNA), the Oxford vaccine uses double-stranded deoxyribonucleic acid (DNA).” -> I suggest they revise these sentences and stress that, while Comirnaty and Moderna are mRNA vaccines, Vaxzevria is an adenovirus vector vaccine. As their sentence suggest in its present form, Vaxzevria could be understood as a DNA vaccine – which is not true.
The correction was made.
Lines 38-43: “While Pfizer-BioNTech (Comirnaty, BioNTech/Pfizer, BNT162b2 Mainz, Germany) and Moderna vaccines (Moderna, mRNA-1273, Massachusetts, EUA) are messenger ribonucleic acid (mRNA) vaccines, the Oxford-AstraZeneca vaccine is an adenovirus vector vaccine [1,4]; contains less ingredients when compared with mRNA vaccines, Table 1, and requires two doses given four weeks apart [3].”
- Line 64 should state exact information on the vaccine!
The reviewer is absolutely right, and we truly appreciate it. The correction was made.
Line 63: “A 33-year-old female was vaccinated with AstraZeneca first vaccine dose (…)
- Again, the figure 1 has to be “professionalized”, e.g. with letters (a – i) for the different panels and equal image sizes in the rows. I suggest the authors assemble all individual pictures into an appropriate figure in Adobe AI, PowerPoint, or a similar program. See PMCID: PMC8447195 (Fig. 1) for example…
We have made it, but in the submission platform it was not possible to upload the original version. We put it here to see the original scientific image that we apply to vaccines.
Lines 71-84: “In that precise moment, an annular plaque with three centimeters of diameter in the infant right leg, was observed below the knee, Figure 1 (A.1). This episode was reported immediately to the pediatrician, who advised the parents to stay alert and vigilant due to mother administration of the AstraZeneca first vaccine dose. After 12-hours, the leg lesion had full recovery and the infant was apyretic. On the 3rd day, two new leg urticarial lesions were observed, one on the back of the right leg (A.2) and the other at anterior region of the left leg (A.3). After 10-hours, both had spontaneously disappeared. On the 4th day after the vaccine, a cutaneous rash was observed in the infant face, with marked uniformly erythematous tender plaques both in the right cheek (B.1) and in the left cheek (B.2). After 11-hours both had extemporaneously vanished. On the 5-day period after the mother vaccine first administration, red marked targetoid cutaneous plaques were seen at the left buttock and the left leg (C.1-C.2). These disappeared after 8h. At the 6-day after the vaccine, right buttock and right leg showed punctual red lesions that vanished also after 12h (D.1-D.2).”

Reviewer 2 Report
The authors took into account all suggested changes.
The manuscript is substantially revised and can be accepted for publication in its present form.
Author Response
No comments were made.
